# Engineered immunological niches to monitor disease activity and treatment efficacy in relapsing multiple sclerosis

Aaron H. Morris [1], Kevin R. Hughes[1], Robert S. Oakes [1], Michelle M. Cai[1], Stephen D. Miller [2], David N. Irani[3] & Lonnie D. Shea [1,4 ✉]

Relapses in multiple sclerosis can result in irreversible nervous system tissue injury. If these events could be detected early, targeted immunotherapy could potentially slow disease progression. We describe the use of engineered biomaterial-based immunological niches amenable to biopsy to provide insights into the phenotype of innate immune cells that control disease activity in a mouse model of multiple sclerosis. Differential gene expression in cells from these niches allow monitoring of disease dynamics and gauging the effectiveness of treatment. A proactive treatment regimen, given in response to signal within the niche but before symptoms appeared, substantially reduced disease. This technology offers a new approach to monitor organ-specific autoimmunity, and represents a platform to analyze immune dysfunction within otherwise inaccessible target tissues.

[1] Department of Biomedical Engineering, University of Michigan, Ann Arbor, MI, USA. [2] Department of Microbiology–Immunology and Interdepartmental Immunobiology Center, Northwestern University Feinberg School of Medicine, Chicago, IL, USA. [3] Department of Neurology, University of Michigan Medical School, Ann Arbor, MI, USA. [4] Department of Chemical Engineering, University of Michigan, Ann Arbor, MI, USA. ✉email: ldshea@umich.edu

Autoimmunity can affect nearly every organ, and these diseases collectively have a prevalence of 15–24 million patients in the United States[1]. Treatment of autoimmune disease before irreversible damage occurs represents a substantial clinical opportunity, yet requires improvements in detecting disease activity and in understanding pathogenic mechanisms. For patients already diagnosed with an organ-specific auto-immune disease, the ability to reliably identify an immunological relapse could fundamentally change treatment approaches. Strategically administered pulses of treatment could obviate the need for lifelong immune suppression. The goal of this investigation is to develop a tool to monitor immune dysregulation occurring within target tissues in order to identify relapse onset and to predict its response to therapy.

Relapsing remitting multiple sclerosis (RRMS) causes accumulating disability due to autoimmune demyelination in the central nervous system (CNS). Sixty percent of patients with RRMS are unable to walk unassisted within 15 years of diagnosis[2]. While a growing arsenal of therapies have been shown to slow relapse rate, new lesion formation on magnetic resonance imaging (MRI) scans of CNS tissues, and disability progression in people with RRMS, an individual's response to one of these therapies can be highly variable[3]. Furthermore, such therapies must be administered continuously over the long-term and frequently cause adverse effects. Finally, while MRI shows areas of demyelination, it provides no information regarding the underlying immunopathology. Prior studies have shown divergent patterns of immune cell infiltration within MS lesions, implying that different immunotherapies could exert variable effects in different patients[4,5]. Clearly, a full characterization of the pathogenic local immune response within the CNS is a major goal, yet remains impossible with current technologies.

Here, we investigate the application of tissue engineering to create easily accessible, subcutaneous immunological niches (IN) that reflect aspects of immune status of CNS tissues in the mouse experimental autoimmune encephalomyelitis (EAE) model of MS. We employ both the adoptive transfer of encephalitogenic T-cells and active immunization models in our studies, with adoptive transfer replicating the effector stage of disease due to infiltration of the transferred T-cells and active immunization including both the induction and effector stages of disease. Porous materials implanted subcutaneously induce cell ingrowth and vascularization, with persistent extravasation of immune cells into the newly forming tissue. We hypothesize that these forming immunological tissues with chronic inflammation can serve as a niche with similar cell infiltrates and phenotypes to other inflammatory sites within the host. Moreover, we hypothesize that biopsy of these engineered IN could detect changes associated with disease activity and could be used to monitor the response to therapy with high reproducibility.

## Results

**EAE alters gene expression at implantable IN**. Microporous poly(ε-caprolactone) (PCL) scaffolds were implanted in the subcutaneous space of SJL mice 2 weeks before adoptively transferring either autoreactive or control T-cells (reactive to PLP$_{139-151}$ or OVA$_{323-339}$, respectively). Through day 7, no mice had symptoms of disease, yet by day 9, all EAE mice became symptomatic (Fig. 1b). INs were harvested and gene expression analyzed via the OpenArray high-throughput gene expression platform on days 7 and 9 (Fig. 1a). Analysis revealed that of the 632 genes analyzed, 130 were differentially expressed between control and diseased mice (time points pooled, $n = 8$, $p < 0.05$) (Supplementary Fig. 1). Of genes that were differentially expressed, a combination of fold change (FC), expression stability, and

elastic net regularization were used to identify 21 genes, one of which, TREM1, was previously unreported in relation to EAE or MS (Supplementary Fig. 2). A radar plot of the log$_2$FC for each of the 21 genes indicated similar patterns of expression between INs isolated from presymptomatic (d7, yellow) and symptomatic (d9, green) EAE mice that differ from those isolated from control mice (pooled time point, dashed) (Fig. 1e). In addition, unsupervised hierarchical clustering demonstrated that control mice and diseased mice cluster separately (Fig. 1d). Collectively, EAE is associated with numerous changes in gene expression at the INs and these changes are detectable before disease symptoms occur.

We next applied computational approaches to develop a scoring system from the 21 gene signature that could distinguish healthy from diseased mice. Two approaches were employed: an unsupervised dimensionality reduction approach, singular value decomposition (SVD); and a supervised machine learning algorithm, bootstrap aggregated decision tree ensemble (Bagged Tree or BT)[6,7]. Results of the SVD and BT were similar and in both cases showed significant differences between control and EAE mice, regardless of time point (Supplementary Fig. 3). These data formed the basis for our trained model for disease classification used throughout the study. A clear separation was observed between INs isolated from diseased and healthy mice by plotting the two scores (Fig. 1f). The signature scores are predictive of disease onset, as the animals at day 7 had no symptoms, yet a high signature score. The score also predicts disease severity, as indicated by the area of each data point in Fig. 1e. Subsequent studies confirmed that healthy mice receiving no adoptive transfer of T-cells, cluster with the OVA controls (Supplementary Fig. 4). Replication with the 21-gene qPCR panel confirmed the OpenArray results and was employed throughout the study (Supplementary Fig. 5).

Follow-up experiments utilized the active immunization model of EAE in SJL mice to demonstrate broader utility of the approach. When compared to mice receiving control immunizations (lacking the PLP$_{139-151}$ peptide) immunized EAE mice demonstrated 222 differentially expressed genes in INs (Supplementary Fig. 6). Time points used for active disease induction were altered to match the slightly slower disease course. The asymptomatic time point was still collected at day 7, but the symptomatic was shifted to day 13 to ensure disease development. Similar to the passive disease induction, a signature of 25 genes was identified that clustered healthy (immunized with no exogenous peptide) control mice together (Supplementary Fig. 7a). Interestingly, four genes overlapped between the signature for the passive and active disease induction, namely: CD163, Il1b, Il1f9, and S100a9. A similar computational scoring system was employed in the immunization model and demonstrated significant differences between control and EAE mice regardless of time point (Supplementary Fig. 7). Again, a clear separation was observed between INs isolated from diseased and healthy mice by plotting the two scores (Supplementary Fig. 7d).

**Characterization of populations within INs and blood**. Cell populations within the INs were subsequently investigated using a panel of eight immune (CD45+) cells commonly involved in EAE[8]. Cells as a percentage of live CD45+ cells were measured, with substantial similarity between healthy and diseased mice, and only minor differences in CD4+ T-cells at day 7 and 21 and CD8+ T-cells at day 9 (Figs. 1c and S8). A similar experiment in the active immunization model of EAE examined cells in INs, blood, and spleens at day 4, 7, and 13 post-immunization. INs demonstrated substantial similarity between healthy and diseased mice at each time point, with the only significant difference being a slight increase in CD11b$^+$ F4/80$^+$ cells at 4-days

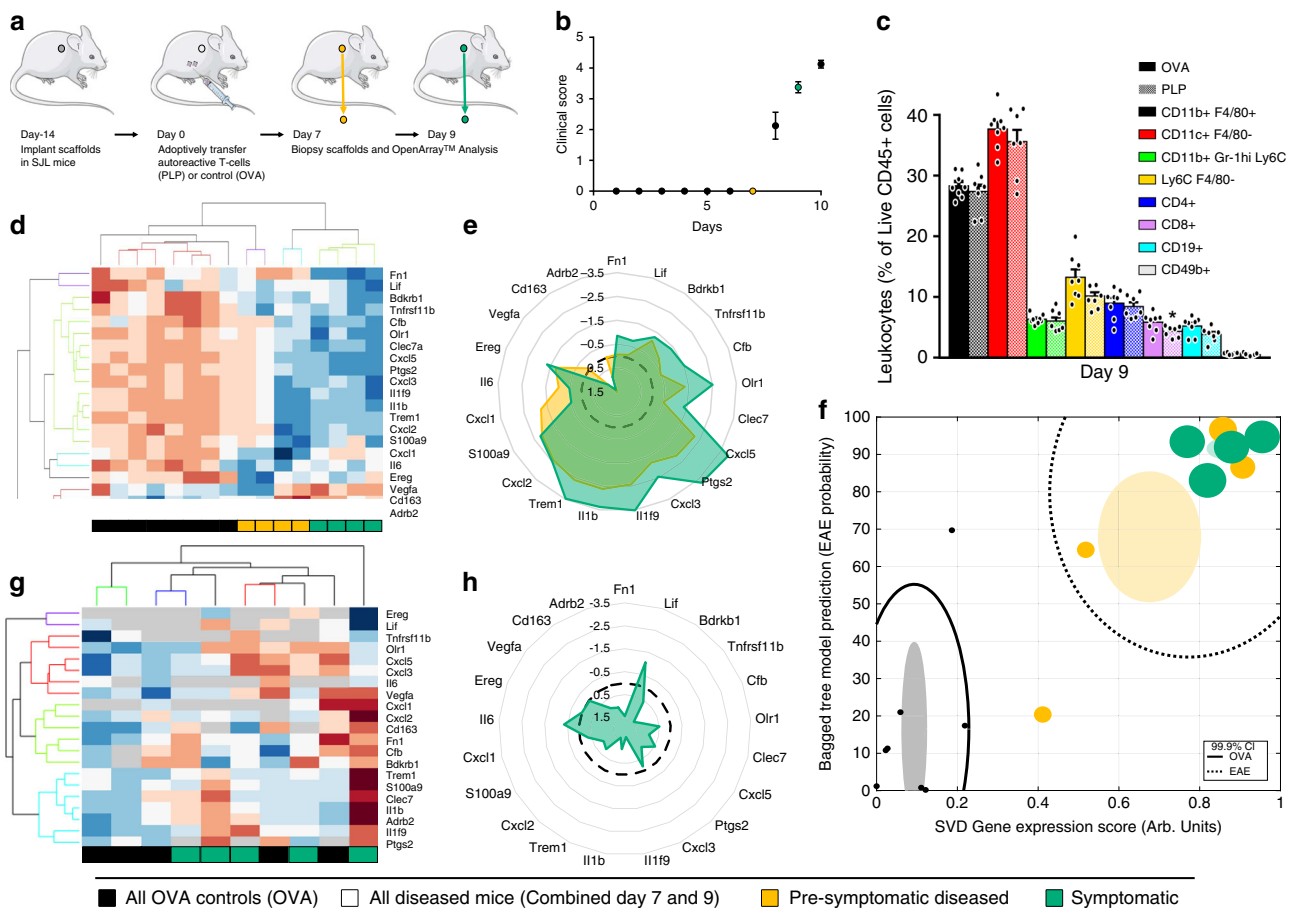

**Fig. 1 IN enables segmenting of diseased and healthy during EAE onset. a** SJL mice were implanted with PCL scaffolds 14 days prior to adoptive transfer of T-cells reactive to either PLP$_{139-151}$ (EAE) or OVA$_{323-339}$ (control). INs were subsequently removed at an asymptomatic (7 day—yellow) or symptomatic (9 day—green) time point for analysis via high-throughput qPCR (OpenArray). **b** Clinical scores of the mice at each time point demonstrate that no mice displayed symptoms of EAE at day 7 post-implantation, but mice were symptomatic by day 9. **c** Flow cytometry for immune cell surface markers demonstrates minor differences between OVA and PLP at day 9. Data are presented as mean + SEM. Statistical analysis performed by two-tailed student's *t* test, asterisk indicates change from control (*$p$ = 0.038). $n$ = 7 (PLP) or 8 (OVA) mice. **d** Heatmap and hierarchical clustering of 21 identified genes of interest, expression levels standardized by gene ($n$ = 8 per condition, 4 at day 7, 4 at day 9). **e** Radar plot demonstrates altered gene expression (as log$_2$FC) for signature genes in presymptomatic (yellow) and symptomatic (green) when compared to time-matched OVA controls (black), with a similar pattern at both time points. **f** Plot of the bagged tree (BT) prediction score versus SVD indicates separation between groups. Black lines indicate 99.9% confidence intervals for pooled diseased or control mice. Filled ovals indicate mean (centroid) and standard error of the mean for each indicated group. Each point is from an individual mouse and increasing point size indicates sicker mice (as an average of the day of explant and day after). **g** Heatmap and hierarchical clustering signature genes from blood (expression levels standardized by gene) indicate no clear clustering of controls from diseased samples, $n$ = 5 per condition. **h** A radar plot of gene expression signature (as log$_2$FC) from blood during symptomatic onset clearly does not show the same patterns as that of the IN. Mouse and syringe cartoon from Servier Medical Art, https://smart.servier.com/smart_image/. Source data for panels **b**–**h** are available as a Source Data file.

post-immunization (Supplementary Fig. 9). Interestingly, the blood and spleen demonstrated increased Ly6C$^+$ F4/80$^-$ inflammatory monocytes in disease at day 13, but the INs do not. Collectively, the large changes in gene expression observed at the IN with disease were likely due to alterations in cell phenotypes within inflammatory microenvironments, rather than cell trafficking. The preponderance of CD45+ cells within the INs are innate immune cells, which are recruited to the CNS during EAE initiation and progression and contribute to damaging the tissue[9–11]. The IN may be sampling the phenotypic alterations of innate immune cells that extravasate into inflammatory environments during disease.

We next examine T-cell subsets within the INs relative to those in spinal cords, inguinal lymph nodes, and spleens because EAE and MS are CD4 T-cell driven diseases[12]. At day 9 after adoptive transfer, five T-cell subsets were measured in each organ: Treg

(CD25$^+$ FoxP3$^+$ CD127$^-$); Th2 (IL-4$^+$); Th1 (IL-17$-$, IFN$\gamma$+); Th1/17 (IL-17+, IFN$\gamma$+); and Th17 (IL-17+, IFN$\gamma$−). INs and spinal cords had similar numbers of T-cells whereas lymph nodes and spleens had an order of magnitude more T-cells. Interestingly, significantly elevated levels of Th1 cells in INs and spinal cords of EAE mice were observed relative to OVA controls (Supplementary Fig. 10). Thus, implanted INs reflect immunological changes in diseased organs that are not reflected in lymphoid tissues.

Next, we tested the hypothesis that the phenotypic alterations captured by analysis of the engineered tissue represent local responses within a tissue that are not captured by analysis of blood. Over the last two decades, numerous studies have examined differential gene expression in the blood of patients with MS and healthy controls[13–17]. Although blood is a rich source of cells, most of these cells are not relevant to the immune

responses occurring within the CNS, which may explain why the overall changes between diseased and healthy are modest and why at least half of differentially expressed genes have few ties to MS pathophysiology. The analysis of blood harvested from symptomatic EAE mice or OVA controls (d10) indicated that the mean expression of signature genes in the blood does not mirror that from the IN, suggesting that INs and blood provide distinct information (Fig. 1h). Unsupervised hierarchical clustering was unable to stratify diseased from healthy mice based on blood data (Fig. 1g); in fact, approximately 18% of samples did not have a valid reading from the blood, with some, such as Ereg, undetectable in most samples.

**INs are dynamic with disease state.** We next investigated the capacity of the IN to monitor disease dynamics that are associated with remission and relapse. Disease was induced by adoptive transfer (cells reactive to PLP$_{139-151}$ or OVA$_{323-339}$) and allowed to progress toward symptomatic remission before IN biopsy at day 21, and relapse was initiated by a second adoptive transfer of cells on day 23 reactive to another immunodominant epitope that simulates epitope spreading during relapse[18] (either PLP$_{178-191}$ or OVA$_{257-264}$). The second IN was biopsied at day 28 (relapse) (schematic and clustergram, Supplementary Fig. 11). Samples isolated from INs during disease remission exhibit INs with similar gene expression to those of control, illustrated by radar plots (Fig. 2a). Furthermore, the cellular makeup of the niches during remission of disease is similar to that during disease onset, thereby implicating cell phenotypes as responsible for observed alterations in gene expression (Supplementary Fig. 8; Fig. 1). During disease relapse, the genes exhibited similar expression to

disease onset, suggesting a return to cell phenotypes associated with disease (Fig. 2b). When analyzed via the trained model, INs from relapsing mice demonstrated high BT and SVD scores, whereas those from animals in remission were similar to control mice (data normalized to time matched controls) (Fig. 2c–f). Importantly, signature scores from INs isolated from remitting mice were significantly reduced compared to mice experiencing disease onset. Relapsing mice had signature scores similar to disease onset. Gene expression at the IN changes during disease onset, reverts toward healthy during remission and indicates disease again during relapse. These findings demonstrate that alterations in disease and signature scores of the INs are dynamic and able to predict disease state.

ROC curves were plotted to determine the diagnostic ability of the system (Supplementary Fig. 12) and demonstrated an AUC of 0.98–1 (depending on metric: SVD, BT, or combined) (95% CI: 0.94–1.03) for the training set, and an AUC of 0.81–0.90 (95% CI: 0.59–1.06) for distinguishing remission and relapse samples, validating the diagnostic efficacy of the signature score derived from INs in diagnosing and prognosing EAE. Collectively, this system offers the opportunity for a patient to be alerted of impending relapse, which would enable pharmacologic intervention to prevent or reduce severity of relapses.

**Preemptive intervention enabled by early detection abrogates disease.** We next tested the hypothesis that pre-emptive intervention enabled by the INs' ability to predict disease onset would improve outcomes. We selected two therapies for investigation: (i) systemic glucocorticoid that represents a standard-of-care treatment and (ii) tolerogenic nanoparticles as an investigational

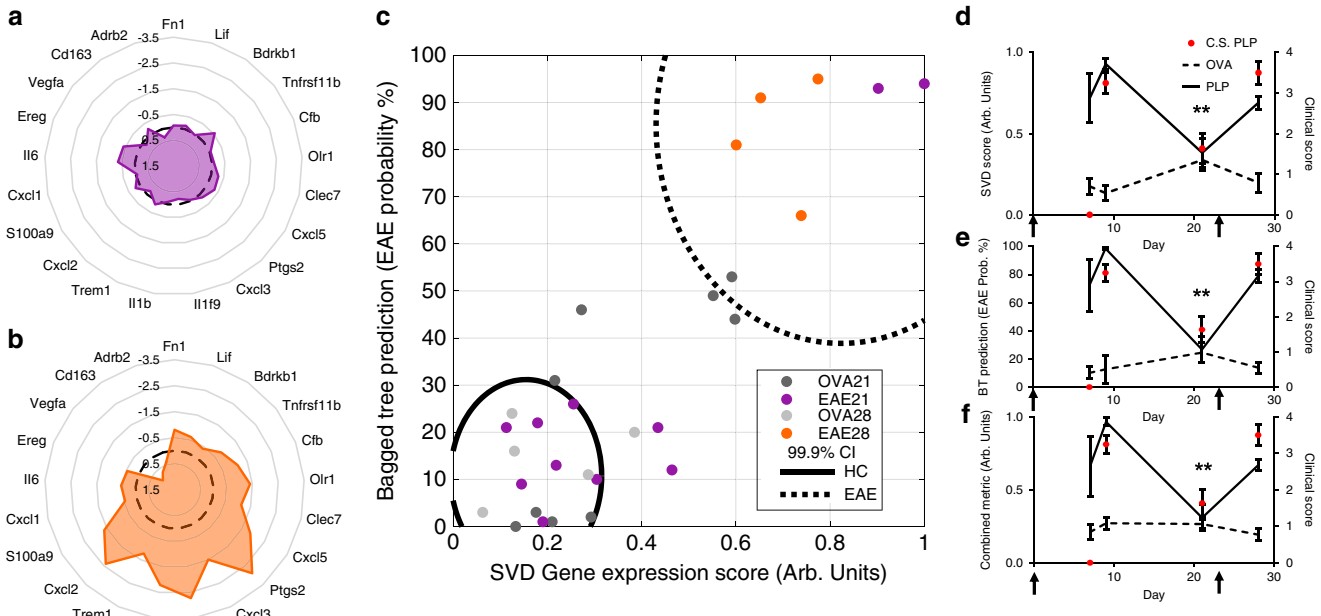

**Fig. 2 Gene signature in IN effectively monitors disease dynamics and indicates relapse.** For relapse studies, INs were implanted subcutaneously 2 weeks before disease induction by adoptive transfer. INs were isolated at day 21 (remission) and day 28 (relapse). Mice received a second adoptive transfer of T-cells (either PLP$_{178-191}$ or OVA$_{257-264}$) on day 23 to induce relapse (indicated on graphs with black arrows). **a, b** Radar plots demonstrate similar gene expression (as log$_2$FC) for signature genes in remission (purple) when compared to time-matched controls (black) (**a**), but altered expression in relapse (orange) (**b**). **c** Plot of BT score versus SVD indicates no separation between healthy and remitting mice, but separation between relapse and healthy. Black lines indicate historic (from disease onset data) 99.9% confidence intervals for pooled diseased or control mice. Each point indicates a single mouse. **d–f** SVD score ($F(3,19) = 5.391$, $p = 0.0074$) (**d**), BT score ($F(3,19) = 9.421$, $p = 0.00050$) (**e**), and a combined metric ($F(3,19) = 6.504$, $p = 0.0033$) (**f**) plotted versus time (left y-axis, black lines) and corresponding clinical scores at each time point (right axis, red points) indicates only the remission t.p. (d21) is significantly different from peak disease (d9). Data are presented as mean ± SEM. A one-way ANOVA was used to compare scores over time for EAE mice. A Bonferroni-corrected post hoc multiple comparisons test was used to compare each time point to symptomatic disease onset (d9). ($n = 9$ OVA at d21, $n = 5$ OVA at d28, $n = 11$ PLP at d21, $n = 4$ PLP at d28). Source data are available as a Source Data file.

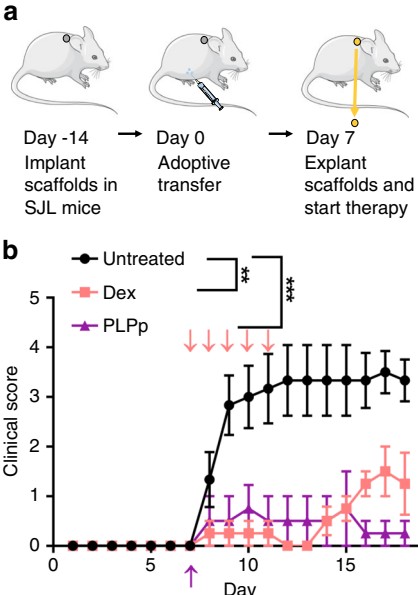

**Fig. 3 Gene signatures enable proactive treatment to prevent disease.**
**a** To test if early detection of disease by INs could enable proactive treatment to prevent the development of EAE. Seven days post-transfer INs were explanted and mice proactively treated with either an I.V. injection of 2.5 mg PLG nanoparticles encapsulating PLP$_{139-151}$ or daily I.P. injections of 5 mg/kg dexamethasone until day 11. **b** Mean (±SEM) clinical scores of the mice at each time point demonstrate that both pre-emptive treatments reduce disease severity when compared to an untreated control. One untreated mouse reached a score of five and had to be euthanized due to humane concerns on day 11. For data presentation and statistical analysis, the final score was propagated for the duration of the experiment. A two-way ANOVA was used to compare scores ($F(2, 11) = 9.973$, $p = 0.0034$). A Dunnett's multiple comparisons test was used to compare each treatment to the untreated control at each time point. ($n = 6$ untreated and 4 treated, **$p \leq 0.0020$ and ***$p \leq 0.00050$ on days 9–18, exact $p$ values in Supplementary Table 1). Pink arrows indicate dexamethasone administration and purple indicate particle administration. Mouse and syringe cartoon from Servier Medical Art, https://smart.servier.com/smart_image/. Source data for panel **b** is available as a Source Data file.

treatment. We have previously reported nanoparticles to induce tolerance in these and similar models, and this platform has recently completed phase IIa human clinical trials for celiac disease (NCT03738475)[19–21]. EAE was induced and INs biopsied for analysis 7 days post-transfer, before delivering therapies: either a daily i.p. injection of 5 mg/kg dexamethasone on days 7–11 or a single i.v. dose of 2.5 mg of antigen encapsulating PLG nanoparticles (described below). Analysis of IN biopsy alerted to immune dysfunction that differed from healthy, and pre-emptive administration of therapy reduced the clinical symptoms and prevented disease onset (Fig. 3; Supplementary Fig. 14). While dexamethasone administered from days 7 to 11 prevented disease onset, the clinical scores began to rise after day 14, yet overall disease burden was significantly reduced through day 18. Interestingly, nanoparticles administered after analysis of the IN at day 7 prevented development of disease through day 18, with only a single administration.

**INs reflect effectiveness of therapy.** The success of tolerogenic particles in mitigating disease led to an analysis of the IN to determine if particle treatment would normalize the scores within the IN, and if the IN could monitor response to therapy a week after administration. IN-implanted mice received an adoptive

transfer of T-cells, and were also injected intravenously with 2.5 mg of antigen encapsulating PLG nanoparticles 2 days post-transfer. Three groups were used: a control group (OVA reactive T-cells and PLP particles), an effective treatment group (PLP reactive T-cells with PLP particles), and an ineffective treatment group (PLP reactive T-cells and OVA particles). INs were biopsied on day 9 and analyzed for the gene signature, which indicated that the effective treatment group had similar signature score and clinical score as the control group. However, the ineffective treatment group had significantly higher signature and clinical scores relative to the control (Fig. 4; Supplementary Fig. 13). The pattern of gene expression in the mice receiving ineffective treatment was similar to the untreated mice during disease onset. To estimate the diagnostic efficacy for treatment monitoring, ROC curves of SVD, BT, and a combined metric were created with AUC values of 0.97–1 (95% CI: 0.89–1.06), suggesting a highly effective treatment monitoring tool (Supplementary Fig. 12).

## Discussion

Autoimmune disease prevalence is on the rise, and although numerous therapies targeting autoimmune conditions have been developed, autoimmunity presents a challenge, because it is typically not diagnosed until substantial damage occurs. We developed an implantable IN that forms a vascularized inflammatory tissue that is dynamic with the status of the immune system. This finding is well supported by reports demonstrating that inflammation surrounding implants is altered by systemic changes associated with various physiological and pathological states, including diabetes, obesity, cancer, and advanced age[22–26]. This implantable biopsy site thus harnesses the host immune system to identify immunological changes within innate immune cells of tissues, which contribute to disease initiation and progression.

The INs contain disease-relevant innate and adaptive immune cells in a tissue that also includes stromal cells and vasculature. The presence of the innate and adaptive immune cells in the INs has similarities to naturally occurring tertiary lymphoid organs (TLOs), which are ectopically formed lymphoid structures at sites of chronic inflammation and are present in a variety of autoimmune diseases[27–29]. TLOs have been associated with biomaterial implants in orthopedic applications, consistent with the role of chronic inflammation at implants and TLOs[30]. Although beyond the scope of this work, the possibility to use INs to generate TLOs for disease monitoring or intervention is intriguing. The INs include a large population of innate immune cells, making them a potentially useful tool for investigating this cellular compartment. Innate immune cells exhibit tremendous alterations during EAE and MS and further investigations into these mechanisms continue to be fruitful in both understanding the basic science and identifying therapeutic targets[31]. For example, myeloid dendritic cells play a key role in EAE progression, particularly in epitope spreading[32]. Another innate immune cell type, inflammatory monocytes, is known to play a critical role in the immunopathology of EAE, as treatments that divert these cells from the CNS to the spleen significantly ameliorate the disease[33,34]. The IN contains large numbers of both DCs and inflammatory monocytes and could possibly be used as a tool to better understand the roles of these cell subsets in EAE and MS pathophysiology. Many of the genes that make up the EAE signatures presented herein are reflective of changing phenotypes of DCs and monocytes. Although the majority of cells in the IN are innate immune cells, adaptive immune cells are also present and there is an opportunity to harness these niches as a source of cells for analysis. For example, the INs could be used to

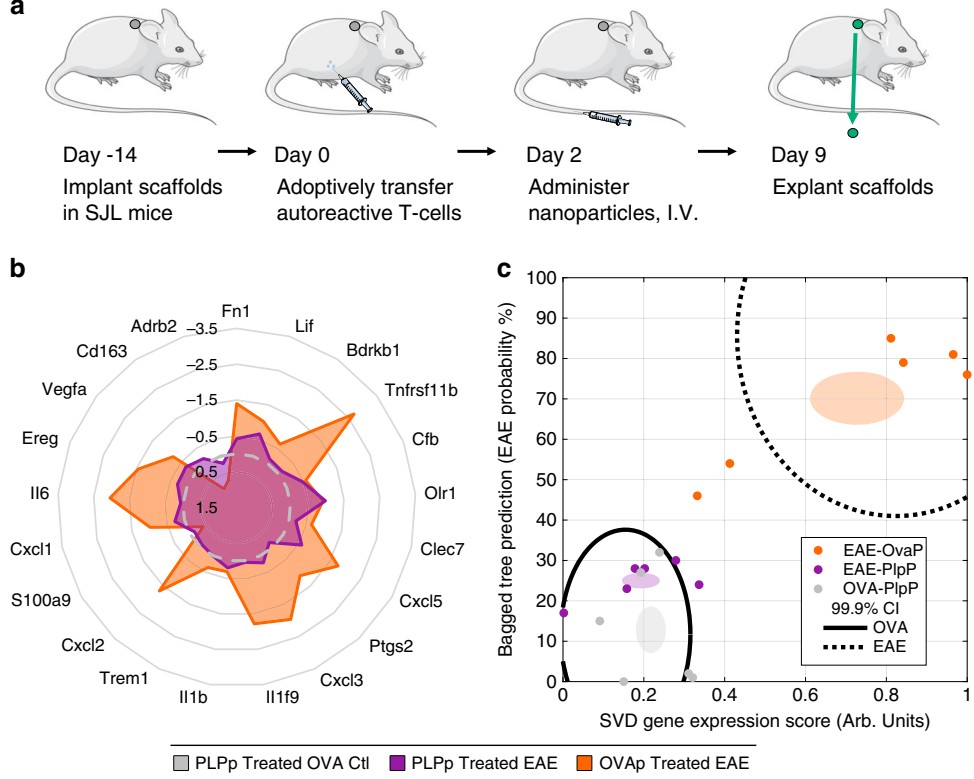

**Fig. 4 Gene signatures indicate response to therapy and enable proactive treatment to prevent disease. a** To test the ability of the IN to indicate response to therapy, 2.5 mg PLG nanoparticles encapsulating either PLP or OVA peptides were administered I.V. 2 days after adoptive transfer and INs were isolated on day 9 (disease onset). **a** Radar plot demonstrating similar gene expression (as log₂FC) for signature genes in EAE mice treated with PLP loaded particles (purple) when compared to time-matched controls treated with particles (grey), but altered expression in EAE mice treated with control (OVA) particles (orange). **b** Plot of BT score versus SVD indicates no separation between healthy mice treated with control particles and EAE mice treated with PLP particles (responders to treatment), but a clear separation in EAE mice treated with OVA particles (nonresponders / control treatment). Black lines indicate historic (from disease onset data) 99.9% confidence intervals for pooled diseased or control mice. Each point indicates a single mouse. Mouse and syringe cartoon from Servier Medical Art, https://smart.servier.com/smart_image/. Source Data for panels **b** and **c** are available as a Source Data file.

develop a further understanding of T-cell subsets in inflamed tissue in EAE by taking longitudinal biopsies throughout the disease course. Certainly, T-cells are responsible for disease initiation, but our understanding continues to evolve[35,36]. Ultimately, the IN assesses peripheral immune events, which the data demonstrates are reflective of immunopathological events occurring in the CNS. Nevertheless, tissue-resident cells of the CNS are absent from the IN and thus, the IN does not fully recapitulate the communication between recruited cells and endogenous cells of the CNS, such as of microglia.

We identified 21 molecular targets in the adoptive transfer model that are differentially expressed at the IN in diseased animals that represent onset or relapse of EAE or MS, enabling interventions to prevent disease development before damage occurs. One of these 21 targets, Trem1, has not previously been associated with EAE or MS, demonstrating the feasibility of harnessing the IN as a tool for preclinical investigations into the mechanisms of disease. Furthermore, the site of the IN is readily accessible and easy to biopsy for analysis. In both human and preclinical models, this site can be easily accessed without detrimental effect to native tissue structures, enabling longitudinal investigations in an individual. In addition to the adoptive transfer model, we developed a signature for the active immunization model of EAE. The signatures from the adoptive transfer and active immunization models contain a subset of overlapping genes, yet also a distinct set of genes owing to differences between

the models and their effects on the immune system. Adoptive transfer models the effector phase of disease, while the immunization model includes the induction and effector stages. Furthermore the immunization model involves the administration of mycotoxin that has dramatic effects on the haemopoietic system including expansion of immature myeloid cells, which may contribute to the observed differences in the gene expression signature[37]. Interestingly, studies investigating T-cell phenotyping in the IN, spinal cord, lymph nodes, and spleen demonstrated that only the IN faithfully recapitulated upregulation of Th1 cells in the spinal cord during EAE onset, highlighting the potential utility of the IN over endogenous lymphoid tissues. Future work may examine engineering these sites to more closely mirror CNS tissues both in architecture and in molecular content. The synthetic nature of these sites could allow engineering with specific factors to investigate the role of the factor in disease initiation and progression[38,39].

Molecular details gleaned from biopsy of the IN enabled preemptive treatment, a strategy which could be transformative for RRMS patients by treating relapses before they occur, with the potential to prevent disease activity altogether. In this report, we tested two treatment strategies for their ability to pre-emptively abrogate disease: pulse glucocorticoids known to benefit MS relapses, and an experimental antigen specific nanoparticle therapy. If administered when the IN indicated disease, the standard of care therapeutic almost completely mitigates disease symptoms

in the short term and reduces overall burden even after cessation of therapy. The emerging, tolerogenic therapy appears to sufficiently induce a longer-term abrogation of disease. Furthermore, molecular insights from the IN can identify the immunological response to the therapies and may serve as a long-term complementary diagnostic. At present, long term disease-modifying treatments for RRMS are typically staged and iterative, involving changing therapies until a patient responds. The IN could be a tool to help monitor treatment efficacy in a more rapid and sensitive manner than either imaging or clinical readouts can provide.

In addition to aiding patient monitoring, the IN is a tool that can provide molecular information on the dysfunctional immune cells within inflamed tissues, and may enable a precision medicine approach for improved subtyping of MS and provide indicators for factors associated with progression[40]. Multiple pathological features can underpin MS in different patients, yet, at present, identifying the subtypes can only occur postmortem. With this IN, molecular information about responses within the tissue could be used to screen individual patients and correlate responses to disease outcomes in addition to therapeutic responses, to enhance prognosis. The IN might also provide a novel approach to study the mechanism of action of both established and investigational disease-modifying therapies for RRMS with proven MRI and clinical benefits. In contrast to blood glucose monitors for Type 1 Diabetes that monitor treatment of established disease, this IN sets the stage for an implantable molecular sensor that continuously reports on the functional status of the immune system to inform disease onset, prognosis, and treatment monitoring.

## Method

**Scaffold fabrication and subcutaneous implantation**. Microporous scaffolds were prepared by mixing PCL with a salt porogen, pressing into molds, polymer sintering, and porogen dissolution[8,41]. All procedures were performed in accordance with the regulations approved by the Animal Care and Use Committee of the University of Michigan. Female SJL/J mice were purchased from Envigo at an age of 6 weeks. Mice were housed in a 12 h light–dark cycle in a pathogen-free environment. Mice were anesthetized with isofluorane, before subcutaneous implantation with scaffolds[42,43]. Mice received subcutaneous injections of carprofen (5 mg/kg) immediately before surgery and 24 h after surgery.

**Disease induction**. Mice were immunized with an emulsion containing 1:1 complete Freund's adjuvant (incomplete Freund's adjuvant [Fisher] with 4 mg/mL heat killed M. tuberculosis H37 Ra [Fisher]): phosphate-buffered saline (PBS) with 2 mg/mL peptide (PLP$_{139-151}$ or OVA$_{323-339}$ [Genscript])[44]. A total volume of 100 μL CFA with peptide was injected subcutaneously at three sites within each mouse. For studies using active induction of EAE the day of immunization was considered day 0 and mice were monitored for disease symptoms after this point. Please note that immunization with mycotoxin has dramatic effects on the haemopoietic system, including expansion of immature myeloid cells[37], which can provide a distinct baseline relative to the adoptive transfer models.

For passive, adoptive transfer disease induction, donor mice were immunized and, 10 days later, donor mice were euthanized and spleens and lymph nodes (inguinal, axillary, and brachial) were harvested before processing into a single cell suspension. Cells were cultured in RPMI with 10% FBS, 2 mM L-glutamine, 100 IU/mL penicillin, 0.1 mg/mL streptomycin, 1× nonessential amino acids solution, 1 mM sodium pyruvate, 10 mM HEPES, and 20 μg/mL peptide antigen. After three days of culture non-adherent cells were harvested, counted, and injected i.p. into recipient mice. For most studies, 30 million cells were injected per mouse, but this inevitably led to some mice becoming moribund, so for remission studies, only 15 million cells were injected. To induce relapse, the same procedure was followed for preparation of cells reactive to either PLP$_{178-191}$ or OVA$_{257-264}$, and 25 million cells were injected i.p. After induction of disease, severity was monitored on a 0–5 scale[21]: 0 = no disease, 1 = hindlimb weakness or limp tail, 2 = hindlimb weakness and limp tail, 3 = partial hindlimb paralysis, 4 = total hindlimb paralysis, 5 = moribund.

**Tissue isolation**. To harvest biopsies of PCL implants for analysis, mice were anesthetized with isoflurane before an incision was made over the surface of the implant. The implant and any adherent encapsulating tissue were pulled through the incision and excised and the incision was closed with sutures. Tissues for RNA

were flash frozen in isopentane on dry ice and stored at −80 °C until analysis. Tissues for flow cytometry were placed into PBS and stored on ice, and tissues for histology were immediately placed in 4% paraformaldehyde. For blood isolation, mice were anesthetized before intracardiac blood draw with EDTA. RBCs were lysed in an ACK solution (Gibco) before washing in PBS. Pellets were resuspended in Trizol and stored at −80 °C until analysis.

**RNA isolation and cDNA synthesis**. RNA was isolated with the Directzol RNA Miniprep kit (Zymo Research) following manufacturer instructions. Samples to be used for OpenArray analysis were also assessed for RNA integrity (RIN) with an RNA fragment analysis with an RNA 6000 Nano Kit (Agilent Technologies), and all samples had a RIN > 8.

cDNA synthesis was performed with the SuperScript™ VILO™ cDNA Synthesis Kit (ThermoFisher Scientific) according to manufacturer instructions. RT was performed with RNA concentrations of 200 ng/μL. Because RNA isolates from blood typically did not achieve this concentration, an RNA clean-up kit was used to increase concentration and purity according to manufacturer instructions (RNA Clean & Concentrator-5, Zymo Research).

**OpenArray high-throughput RT-qPCR**. For high-throughput gene expression analysis, OpenArray panels focused on mouse inflammatory pathways were used. Panels and accompanying reagents were purchased from ThermoFisher (Applied Biosystems™ TaqMan™ OA Mouse Inflammation Panel, Cat. No. 4475393). The panels contained 632 validated genes that have known roles in inflammation as well as 16 endogenous controls (housekeeping genes). cDNA was prepared as described above and OA analysis and quality control was performed on a fee-for-service basis by the UM DNA Sequencing core. The core uses a robotic OA AccuFill system and the QuantStudio 12k Flex RT-PCR system (ThermoFisher Scientific). For OpenArray of adoptive transfer samples a total of $n = 8$ samples per condition (diseased or healthy) were analyzed with 4 from each time point (day 7 or 9). Each sample was from an independent mouse and samples alternated between healthy and diseased in their placement on the chip to minimize spatial bias. For immunization studies, two time points were used for OpenArray, day 7 or 13. A total of $n = 8$ immunized control (4 from each time point) and $n = 16$ diseased (8 from each time point) were used.

**Selection of genes of interest**. Samples were analyzed first to remove genes that did not have readable data in more than one of the four mice in each condition. Next, any samples that were missing data were filled with the median for the overall dataset. This was required because downstream analysis requires complete matrices. Ultimately, 500 genes of the 648 were used for this study. Each of the 16 reference genes were analyzed via the NormFinder algorithm to select the three most stable reference genes: Hmbs, Polr2a, and Ubc. $\Delta C_q$ values were calculated for each gene from the average of the reference genes for that sample. Next, FC, $p$ values, and prediction scores derived from elastic net regularization (MATLAB's lasso function, $\alpha = 0.01$, leave one out cross-validation) were calculated for each gene using time-matched controls[44]. Box-plots show log$_2$FC, centered on the median of the time matched healthy controls. All identified genes of interest were identified as predictors (elastic net scores > 0), had FC > 1.5, and $p < 0.05$. In addition, we attempted to include genes that increased and decreased during disease to make the model more robust and ease the eventual construction of a sensor.

Once these genes of interest were identified, all other studies were conducted via RT-qPCR analysis in 384-well plates. The same TaqMan probes used in the OpenArray chips were ordered from ThermoFisher Scientific and samples were alternated between healthy and diseased in their placement on the plate. TaqMan Gene Expression Mastermix was used and a final volume of 9 μL was used for each well of the 384-well plate. RT-qPCR was performed on the QuantStudio ViiA 7 system and $C_q$ values determined by the accompanying software. Non-detects were left blank for any statistical analyses, but were filled with the median of all samples for SVD (because it requires complete matrices). Samples from the original OpenArray data were run on every 384-well plate to allow for a correction factor to be applied to account for variability.

**Gene signature scores and analysis**. Firstly, unsupervised hierarchical clustering analysis was performed using MATLAB's clustergram tool which plots dendrograms to indicate samples and genes that cluster together. Ultimately, genes of interest were identified, and computational approaches were used to create two metrics for evaluating whether mice were sick or healthy. SVD using MATLAB's svds function was applied to create a gene signature score using an unsupervised technique. Next, a supervised machine learning approach bootstrap aggregated decision tree ensemble (Bagged Tree) was trained to classify samples as healthy or diseased. We employed MATLAB's fitcensemble function with the Bag method.

**Histology**. For histological analysis, samples were excised and stored immediately in 4% PFA overnight to fix the tissue before being bisected and transferred to the University of Michigan In Vivo Animal Core for sectioning and staining with hematoxylin and eosin using standard protocols. Images were taken at 20× within the thickness of the IN.

**Flow cytometry**. INs from adoptively transferred animals were prepared for flow cytometry by mechanical dissection and enzymatic incubation[8,45]. Briefly, samples were minced with a scalpel and incubated for 20 min in Liberase TL (Roche) at 37 °C. INs were then mashed through a 70 μm filter which was washed extensively with FACS buffer: PBS (Life Technologies) with 0.5% bovine serum albumin (Sigma Aldrich) and 2 mM EDTA (Gibco). Cells were equally split into two tubes to enable staining and analysis of innate and adaptive immune cells from the same IN and then blocked with anti-CD16/32 (1:50, clone 93, eBioscience). Each tube was stained with Live/Dead Fixable Red (Life Technologies) and Alexa Fluor® 700 anti-CD45 (1:125, clone 30-F11, Biolegend). The adaptive immune panel was also stained with: FITC anti-CD8 (1:25, clone 53-6.7, Biolegend), Pacific Blue™ anti-CD19 (1:100, clone 6D5, Biolegend), PE-Cy7 anti-CD49b (1:30, clone DX5, Biolegend), and V500 anti-CD4 (1:100, clone RM4-5, BD Biosciences). The innate immune panel was also stained with: APC anti-CD11c (1:80, clone N418, Biolegend), FITC anti-Ly6C (1:100, clone HK.14, Biolegend), Pacific Blue™ anti- Ly-6G/Ly-6C (Gr-1) (1:70, clone RB6-8C5, Biolegend), PE-Cy7 anti-F4/80 (1:80, clone BM8, Biolegend), and V500 anti-CD11b (1:100, clone M1/70, BD Biosciences). Samples were analyzed on a Cytoflex Cell Analyzer, and all single-color controls and FMOs were used to aid with gating and compensation (Supplementary Figs. 15 and 16), with subsequent data analysis on FlowJo (v.X).

INs, blood, and spleens from immunized mice were analyzed similarly to above. Blood was collected via cardiac puncture, and a single cell splenocyte homogenate was obtained by mashing through a 70 μm filter which was washed extensively with FACS buffer. Red blood cells were then lysed with ACK lysis buffer (Gibco). The panel above was used for analysis, except no anti-CD49b antibody was used.

For analysis of T-cell subsets, spinal cords and INs were harvested and processed by mincing and liberase digestion as described above. Lymph nodes and spleens were mashed through a 70 μm filter. Samples were divided for analysis with two staining panels: Treg and Th panels. Both panels were stained with Live/Dead Fixable Violet (Life Technologies), BV510 anti-CD3 (1:80, clone 17A2, Biolegend), and FITC anti-CD4 (1:125, clone RM4-5, Biolegend). The Treg panel added PE anti-CD25 (1:80, clone PC61, Biolegend), APC anti-FoxP3 (1:8, clone FJK-16s, eBioscience), and PE-Cy7 anti-CD127 (1:80, clone A7R34, Biolegend). The Th panel added PE anti-IL-17a (1:16, clone TC11-18H10.1, Biolegend), APC anti-IL-4 (1:16, clone 11B11, Biolegend), and PE-Cy7 anti-IFNγ (1:16, clone XMG1.2, Biolegend). After processing and staining for surface markers as described above, cells were fixed and permeabilized using the eBioscience Foxp3/Transcription Factor Staining Buffer Set according to manufacturer instructions, before intracellular stains. For gating scheme, see Supplementary Figs. 17 and 18.

**Nanoparticle fabrication and administration**. Peptide encapsulating nanoparticles were fabricated with the double emulsion method[46]. Briefly, 150 μL of antigen dissolved in PBS (50 mg/mL $PLP_{139-151}$ or $OVA_{323-339}$ [Genscript]) was added to 2 mL of 20% w/v poly(lactide-co-glycolide) (inherent viscosity: 0.17 dL/g, Lactel). This solution was sonicated for 30 s before the addition of 10 mL of 1% w/v poly(ethylene-alt-maleic anhydride) in water. This was again sonicated to form the double emulsion which was poured into 200 mL of 0.5% w/v poly(ethylene-alt-maleic anhydride) and stirred overnight. Particles were washed extensively before lyophilization in cryoprotectant. The size and zeta potential of the particles were determined by dynamic light scattering (DLS) by mixing 10 mL of a 25 mg/mL particle solution into 990 mL of MilliQ water using a Malvern Zetasizer ZSP.

For treatment studies, mice were implanted with INs and 14 days later adoptively transferred with 30 million T-cells. Two days, post-transfer, mice received a single bolus i.v. injection of 2.5 mg $PLP_{139-151}$ or $OVA_{323-339}$ loaded nanoparticles. Mice were monitored daily for symptoms of EAE, and INs were removed on day 9.

**Statistics**. All statistics were calculated with MATLAB or GraphPad Prism software. When comparing pooled control vs diseased mice, two-way ANOVA was used to determine significance. A post hoc multiple comparisons with a Bonferroni correction was then used to compare each time point to its time-matched control. When comparing the scores of EAE mice over time a one-way ANOVA with a Bonferroni corrected post-hoc multiple comparisons test was used. In addition, the same method was used to compare the three particle treated groups at a single time point. Student's $t$ tests were used to analyze flow cytometry data as comparisons were made between healthy and diseases within each time point (no comparisons over time). ROC curves were plotted and analyzed in GraphPad. Box plots show the median, 25–75th percentiles and most extreme data points not considered outliers (outliers are indicated by red+).

**Reporting summary**. Further information on research design is available in the Nature Research Reporting Summary linked to this article.

## Data availability

To increase transparency, all OpenArray Datasets (Cq values) are also provided within source data including: adoptive transfers, immunizations, and healthy controls. Any other data is available from the corresponding author upon reasonable request. Source data are provided with this paper.

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

## Acknowledgements

We would like to thank the UM Precision Health Scholars grant for funding this work. This work was supported in part by NIH grants R01AI148076, R01EB013198, R01DK121462, R21AI147677, and R01CA243916. A.H.M. is supported by a Michigan Life Science Fellows award and a NIH T32 grant DE007057-43 and would like to thank both of these funders for their support. We would like to thank Christopher Krebs, PhD and the University of Michigan DNA Sequencing Core for assistance in OpenArray processing and analysis. We would also like to thank the University of Michigan Flow Cytometry Core and In Vivo Animal Core for flow cytometry and histology resources, respectively. Thank you to Servier Medical Art for providing the cartoons of the mouse and syringe (modified and used in this paper) under a Creative Commons license: https://creativecommons.org/licenses/by/3.0/. Finally, we would like to thank Dr. Bethany Moore for helpful conversations and insights.

## Author contributions

A.H.M., K.R.H., R.S.O., and L.D.S. conceived aspects of project. A.H.M., K.R.H., R.S.O., and M.M.C. performed the experiments and analyzed the data. A.H.M., K.R.H., R.S.O., and L.D.S. designed the experiments. D.N.I., S.D.M., and L.D.S. helped analyze and interpret the data and contributed to the supervision of the project. A.H.M., K.R.H., and R.S.O. visualized the data and generated the display items. A.H.M., K.R.H., and L.D.S. drafted the original paper. A.H.M., K.R.H., R.S.O., M.M.C., D.N.I., S.D.M., and L.D.S. edited and provided input into all aspects of the paper. A.H.M., S.D.M., and L.D.S. acquired funding to support this project.

## Competing interests

This paper uses nanoparticles for which L.D.S. and S.D.M. have an interest in Cour Pharmaceutical, who have licensed the technology. L.D.S., A.H.M., R.S.O., and K.R.H. are co-inventors on pending patent applications related to engineered immunological niches. The authors declare no further competing interests.
