## [Peer Review File · Nature Communications]

Reviewers' comments:

Reviewer #1 (Remarks to the Author):

The manuscript provides an interesting approach to assess immune regulation in experimental autoimmune encephalomyelitis (EAE), a model of multiple sclerosis (MS). There are certain limitations of the study that can be possibly overcome by argumentation and additional data generation. Firstly, the immune response in EAE is leading to tissue damage in the central nervous system (CNS). The niche model does not provide the same compartment as the CNS and does not provide a blood-brain-barrier (BBB). Therefore, it can be considered rather as a tool to assess the immune events in draining lymph nodes. Of course, such a tool to be implanted in MS patients would be of great value for assessment of immunity and relapse-preceding events. Secondly, the model that is used in the study is induction of EAE in SJL/J mice that are transferred with PLP178-191 specific T cells. This is a good model for assessment of T cell immunology in EAE. It is not such a good model for assessing additional contributions of the immune system to induction of CNS lesions like assessment of the B cell compartment in EAE. Therefore, additional models should be searched, established and evaluated that provide the possibility for assessment of the more complex immune regulation of EAE and MS.

Specific points:

The introduction is too general. It would be of interest to know why the authors used the selected model.

In MS research understanding relapse and chronicity is of major importance. Therefore, in the results part the authors should provide better description of the findings in this phase of the disease.

The discussion should be also more focused on the findings and provide some assessment what is novel about the findings regarding understanding immunity in EAE and MS. Is there anything novel that cannot be found by assessment of lymphoid tissues in EAE? Please compare the findings with findings in the literature regarding the relapsing and chronic phases of EAE. Please provide the rationale why the approach would be still of great value for use in humans.

Reviewer #2 (Remarks to the Author):

In the manuscript "Engineered immunological niches to monitor disease activity and treatment efficacy in relapsing multiple sclerosis" authored by Dr. Shea and colleagues, the authors describe a novel methodology to investigate the profile of the peripheral immune cells and correlate it with disease activity in experimental autoimmune encephalomyelitis (EAE), model of MS. According to the authors, this novel technology, which relies on subcutaneous injection of a porous scaffold matrix, offers a new approach to investigate the immune system and proactively identify a treatment window to suppress relapses in EAE and other autoimmune diseases. Although some of the data are not referenced in the manuscript, the overall amount of data is convincing; and the use of a machine learning approach to predict relapses in SJL EAE model is novel and very interesting. Given that MS is a debilitating disease that affects 2.5 million people worldwide and that treatments have had limited efficacy, the use of such a technology would greatly advance the field and shed light on novel molecules that can be targeted to treat MS as well as it can be used to predict relapses. Therefore, there is enthusiasm for this study. However, lack of a more detailed analysis of the cellular composition of leukocytes in niches, an unusual animal model (passive instead of active EAE) for these analyses, and lack of rigor in data presentation undermine the manuscript quality. A summarized list of deficiencies is found below:

1. In fig. 1, authors screen the niches for leukocytes involved in EAE pathogenesis and conclude that there is a preponderance of innate cells over adaptive cells. However, EAE is known to be a T cell-dependent model. Authors should have better characterized the cellular subpopulations (Th17, Th1, ThG) of T cells and other cell types to have a better assessment of the inflammation that predicts relapses, and then followed the dynamics of such cell populations in different stages of actively induced EAE.

2. It is not clear why authors resourced to passive EAE instead of active EAE, given that active immunization would result in RR EAE, which would provide more insight into the dynamics of the immune response over the course of the disease.
3. Throughout the manuscript, figures are not referenced in the order they appear in the text. Some of them are not referenced at all (e.g. Fig. 2C, S. Fig 11 and S. Fig. 12).
4. Did authors compare the cellular composition in niches with those in CNS and dLN? This piece of information could be important to strengthen their model.
5. For the preemptive intervention experiments, active immunization would be a better model than passive EAE given the natural course of the disease.
6. Still on the preemptive experiments: authors utilized an Ag-specific tolerance induction treatment to test their hypothesis. However, wouldn't it be better to test the effect of FDA-approved drugs for MS instead as it is more relevant from a clinical perspective? And, according to their hypothesis that probing the niches at different time-points would provide data that could predict a relapse, wouldn't a better proof-of-concept experiment rely on treating mice at different time-points according to the data provided from the niches and check for relapses?
6. Discussion is too short and lacks depth.
7. Fix figure legend for S. Fig 3.
8. Stainings on Fig. S11 and S12 are not convincing: (i) CD11b+F4/80- does not seem realistic, (ii) gates for CD11b+F4/80+, Ly6c+F4/80+, CD11b+Gr1+Ly6c-, CD19+CD49b+ are cutting through the negative population, (iii) plot CD11c+F4/80+, staining is not compensated.

We would like to thank the reviewers for their thoughtful review of the manuscript. We were fortunate to have been able to perform several experiments in parallel to address the below concerns before COVID-19 shut down the labs at the University of Michigan. We have provided an item by item response to the concerns below, and believe the manuscript has substantially improved with the revisions.

Reviewer #1

1. The niche model does not provide the same compartment as the CNS and does not provide a blood-brain-barrier (BBB). Therefore, it can be considered rather as a tool to assess the immune events in draining lymph nodes. Of course, such a tool to be implanted in MS patients would be of great value for assessment of immunity and relapse-preceding events. Secondly, the model that is used in the study is induction of EAE in SJL/J mice that are transferred with PLP178-191 specific T cells...

We thank the Reviewer for the supportive comments and excitement in using the niche as a tool for assessing immunity. We agree that it is a tool for assessing immune events, yet is not a perfect representation of the CNS.

*We agree with the Reviewer that investigations into the active immunization model would strengthen the manuscript. As such we have included experiments with active immunization for EAE onset and signature identification (Fig. S6-7). Furthermore, we characterized immune cell populations in the IN, blood, and spleens during actively induced disease at three time points (Fig. S9). We are pleased to report that this approach extends more broadly to actively induced disease. These models have several commonalities, yet also differences, particularly due to the presence of the adjuvant in the active model. Regarding the commonalities, four genes in the signatures overlapped between active and passively induced disease. *Il1b*, *Il1f9*, *S100a9*, and *CD163* are conserved between the two models, with each of these factors known to be produced by activated monocytic cells and neutrophils that play a large role in orchestrating immune responses in both the active and passive models EAE.*

2. The introduction is too general. It would be of interest to know why the authors used the selected model.

Thank you for the comment. Regarding the model selection, the primary reason for choosing the adoptive transfer model was to target the effector stage of disease in which the transferred T cells begin infiltration shortly after adoptive transfer. We hypothesized that changes measured in the adoptive transfer model would be more directly related to disease without the off-target responses to the Mycobacterium and TLR signaling induced in active immunization models that would not be present in humans. To exemplify the potential for off target response, the niches are predominantly composed of innate immune cells, and CFA (used for active EAE) is known to have dramatic effects on the haemopoietic system including “expansion of Mac-1+ immature myeloid cells” (Billiau and Matthys, Journal of Leukocyte Biology, 2001).

Nevertheless, we agree with the Reviewer that investigations into the active immunization model would strengthen the manuscript. The active model involves delivery of an adjuvant with the antigen and thus involves both disease induction and effector stages of disease. We have now repeated the EAE onset experiments and signature identification in the active immunization model of EAE (Fig. S6-7). Furthermore, we characterized immune cell populations in the IN, blood, and spleens during actively induced disease at three time points (Fig. S9). We are pleased to report that this approach extends more broadly to actively induced disease. Interestingly, four genes in the signatures overlap between active and passively induced disease.

Given the inclusion of both active and passive models, we have added text to clarify the advantages of each model, and the similarities and differences of each. This text can be found on pages 6-7.

We have also added more detail to the introduction and the revised text can be found on page 1-2.

3. In MS research understanding relapse and chronicity is of major importance. Therefore, in the results part the authors should provide better description of the findings in this phase of the disease.

Thank you for this comment. We agree and have attempted to highlight the results by providing more thorough descriptions. We have added several sentences discussing the results the relapsing and remitting phases. The adjusted text (below) can be found on page 4 of the revised manuscript.

“... Samples isolated during disease remission exhibit INs with similar gene expression to those of control, illustrated by radar plots (Fig. 2a). Furthermore, the cellular makeup of the niches during remission of disease is similar to that during disease onset, thereby implicating cell phenotypes as responsible for observed alterations in gene expression (Fig. S8 and Figure 1). During disease relapse, the genes exhibited similar expression to disease onset, suggesting a return to cell phenotypes associated with disease (Fig 2b). When analyzed via the trained model, INs from relapsing mice demonstrated high BT and SVD scores, whereas those from animals in remission were similar to control mice (data normalized to time matched controls) (Fig 2c-f). Importantly, signature scores from INs isolated from remitting mice were significantly reduced compared to mice experiencing disease onset. Relapsing mice had signature scores similar to disease onset. Gene expression at the IN changes during disease onset, reverts toward healthy during remission and indicates disease again during relapse. These findings demonstrate that alterations in disease and signature scores of the INs are dynamic and able to predict disease state. “

4. The discussion should be also more focused on the findings and provide some assessment what is novel about the findings regarding understanding immunity in EAE and MS. Is there anything **novel that cannot be found by assessment of lymphoid tissues in EAE**? Please compare the findings with findings in the literature regarding the relapsing and chronic phases of EAE. Please provide the rationale why the approach would be still of great value for use in humans.

Thank you for this point. We have added substantially to the discussion including context regarding the relapsing phases of EAE. This includes the addition of thirteen

more references highlighting important work in the field (25-37). We also include further discussion of the inclusion of TREM1 as a gene in our signature. TREM1 has not been shown to play a role in EAE or MS, but has a known role in RA. Our approach enabled identification of TREM1 as a candidate biomarker in EAE using our accessible IN. In addition, as part of the revision we examined immune populations in INs as compared to spleens in active immunization. This clearly demonstrates the difference in composition between the IN being largely innate and lymphoid tissues largely adaptive immune cells (Fig. S9).

We hypothesize that the IN samples innate immune cells in an unbiased way. The enrichment of these innate cells [the IN contains more innate immune cells ($\approx 75-80\%$) than adaptive cells ($\approx 20-25\%$)] may provide the capacity to detect changes in innate immune cell function and phenotype that is associated with disease. The INs enable examination of phenotypic changes that occur in innate immune cells that have infiltrated a tissue, which may better represent disease-relevant phenotypes than those in the blood.

In the adoptive transfer model, we characterized T-cell subsets in INs, inguinal lymph nodes, spleen, and CNS and only the IN was able to faithfully recapitulate the increase in Th1 cells observed in the CNS during disease (Fig S10). This demonstrates utility of the IN over lymphoid tissues in some scenarios.

In addition to recapitulating key aspects of the innate and adaptive response in disease, the IN is accessible in the subcutaneous space and can be biopsied without harming native tissue structures. We show that the INs detect changes in inflammatory status during disease onset and relapse that can be reset by the administration of tolerogenic nanoparticles. Clearly, the dynamic response of these niches enables the monitoring of both disease state, but also response to treatment by interrogating a synthetically engineered tissue. The engineered site allows longitudinal monitoring in tissues derived from a single host without explanting organs or euthanasia. In this regard it would be enabling in humans. Not only could it have tremendous advantages for diagnosis, but it could readily be used for basic immunology studies in a human host.

Reviewer #2

1. In fig. 1, authors screen the niches for leukocytes involved in EAE pathogenesis and conclude that there is a preponderance of innate cells over adaptive cells. However, EAE is known to be a T cell-dependent model. Authors should have better characterized the cellular subpopulations (Th17, Th1, ThG) of T cells...

We thank the reviewer for this insightful point. Because the IN contains more innate immune cells ($\approx 75-80\%$) than adaptive cells ($\approx 20-25\%$), we had elected to focus our attention the innate cells as the large numbers provide a more thorough analysis based on the ability to adequately sample a representative population. However, we also agree that examination of T-cell subsets makes for a more complete picture of disease. We examined T-cell subsets by flow cytometry in the INs, spinal cords, inguinal lymph nodes, and spleens of mice receiving adoptive transfers of encephalitogenic T-cells. Collectively, INs were the only tissue to faithfully recapitulate the increase in Th1 cells observed in the spinal cord at day 9 (Fig. S10).

2. It is not clear why authors resourced to passive EAE instead of active EAE, given that active immunization would result in RR EAE, which would provide more insight into the dynamics of the immune response over the course of the disease.

*Thank you for the comment. Regarding the model selection, the primary reason for choosing the adoptive transfer model was to target the effector stage of disease in which the transferred T cells begin infiltration shortly after adoptive transfer. We hypothesized that changes measured in the adoptive transfer model would be more directly related to disease without the off-target responses to the Mycobacterium and TLR signaling induced in immunization models that would not be present in humans. To exemplify the potential for off target response, the niches are predominantly composed of innate immune cells, and CFA (used for active EAE) is known to have dramatic effects on the haemopoietic system including “expansion of Mac-1+ immature myeloid cells” (Billiau and Matthys, *Journal of Leukocyte Biology*, 2001).*

Nevertheless, we agree with the Reviewer that investigations into the active immunization model would strengthen the manuscript. The active model involves delivery of an adjuvant with the antigen and thus involves both disease induction and effector stages of disease. We have repeated the EAE onset experiments and signature identification in the active immunization model of EAE (Fig. S6-7). Furthermore, we characterized immune cell populations in the IN, blood, and spleens during actively induced disease at three time points (Fig. S9). We are pleased to report that this approach extends more broadly to actively induced disease. Interestingly, four genes in the signatures overlap between active and passively induced disease.

Given the inclusion of both active and passive models, we have added text to clarify the advantages of each model, and the similarities and differences of each. This text can be found on pages 6-7.

We have also added more detail to the introduction and the revised text can be found on page 1-2.

3. Throughout the manuscript, figures are not referenced in the order they appear in the text. Some of them are not referenced at all (e.g. Fig. 2C, S. Fig 11 and S. Fig. 12).

We thank the Reviewer for noticing this oversight and have changed the order of figures and ensured each figure is referenced within the manuscript.

4. Did authors compare the cellular composition in niches with those in CNS and dLN? This piece of information could be important to strengthen their model.

We thank the reviewer for this suggestion and have performed two studies to satisfy this point. Firstly, in the T-cell subpopulation experiments described above we examined T-cell subsets in INs, CNS, dLN, and spleens (Fig. S10). Additionally, for the active immunization model suggested by the reviewer, we examined INs, blood, and spleens at three different time points (Fig. S9). The INs faithfully represent changes in T-cell subsets that are present in the CNS during disease. Overall, the INs present somewhat similar patterns to the dLNs and spleens, but importantly enrich innate immune cells. The preponderance of immune cells in the IN are myeloid,

whereas in dLN and spleens they are lymphoid. Enriching innate immune cells provides a critical advantage in monitoring EAE as treatments to divert inflammatory monocytes away from CNS ameliorate EAE showing that peripherally-derived myeloid cells play a major role in the CNS immunopathology (Getts 2014).

5. For the preemptive intervention experiments, active immunization would be a better model than passive EAE given the natural course of the disease.

We performed two sets of experiments examining treatment of the animals before the onset of symptoms. We selected the adoptive transfer model in both cases, because we believe this model avoids the confounding effects of the adjuvant to assess treatment efficacy. Although the disease course is similar for passive and active EAE, the passive model results in an animal replete with highly encephalitogenic T-cells that can invade the CNS nearly immediately. Thus, the animal has fully established disease-potential. We wanted to demonstrate that treatment would be effective in stopping the effector phase of the disease not modulating the induction.

We have modified the manuscript and added a new figure (Fig. 4) to clarify that we tested two pre-emptive intervention experiments, and have included a brief description here. In the first experiment, we sought to identify whether the early warning system of the IN would enable intervention to prevent the development of disease. As such we explanted scaffolds at the presymptomatic stages and intervened with therapies in mice that would develop disease. The two therapies chosen were: i) dexamethasone injected daily starting on day 7 and ii) Ag-specific tolerogenic therapies injected once on day 7. We then monitored the mice for symptoms (Fig. 3).

In the second experiment, we sought to determine if we could measure the treatment efficacy induced by tolerogenic nanoparticles. For this experiment we treated mice with particles once on day 2 following adoptive transfer. We then examined the signal at the IN one week later to determine if we could use the IN to measure the response to an ineffective vs effective treatment (Fig. 4).

6. Still on the preemptive experiments: authors utilized an Ag-specific tolerance induction treatment to test their hypothesis. However, wouldn't it be better to test the effect of FDA-approved drugs for MS instead as it is more relevant from a clinical perspective? And, according to their hypothesis that probing the niches at different time-points would provide data that could predict a relapse, wouldn't a better proof-of-concept experiment rely on treating mice at different time-points according to the data provided from the niches and check for relapses?

For pre-emptive intervention (Fig 3), we analyzed niches at day 7 to determine disease onset and then intervened with a glucocorticoid (standard-of-care), antigen-specific nanoparticle treatment, or untreated control. The pre-emptive treatments nearly completely abolished the disease.

Our second experiment (Fig 4) measured the ability of the niche to measure treatment response. For this, we chose a model of effective (PLP-loaded) or ineffective (OVA-loaded) treatment which we implemented at day 2 and measured response both through clinical scores and also the signature from the IN. In addition to providing a platform to monitor treatment response, the tolerogenic particle therapies are an

emerging treatment that will soon be clinically available. Cour Pharmaceuticals has recently finished Phase 2 clinical trials using this platform for celiac disease with great success. Moreover, Cour has submitted an IND to the FDA to extend the platform for treating MS by encapsulating a cocktail of encephalitogenic epitopes from myelin. We have reorganized the presentation of this data to clarify the study objectives and results.

7. Discussion is too short and lacks depth.

Thank you for this comment. We have expanded the discussion extensively to more thoroughly contextualize the literature (25-37). We discuss implications of the work and potentially key cell types. We also include discussion of the inclusion of TREM1 as a gene in our signature. TREM1 has not been shown to play a role in EAE or MS, but has a known role in rheumatoid arthritis (RA). Our approach enabled identification of TREM1 as a candidate biomarker in EAE using our accessible IN. Finally, we discuss the ability to leverage this novel tool in the human setting. We believe that these additions substantially strengthen the manuscript.

8. Fix figure legend for S. Fig 3.

Thank you for catching this mistake. We have corrected the figure legend.

9. Stainings on Fig. S11 and S12 are not convincing: (i) CD11b+F480- does not seem realistic, (ii) gates for CD11b+F4/80+, Ly6c+F4/80+, CD11b+Gr1+Ly6c-, CD19+CD49b+ are cutting through the negative population, (iii) plot CD11c+F4/80+, staining is not compensated.

Thank you for the suggestions. We gated our studies using FMO stains, but at the Reviewer's suggestion have re-examined and gated our flow experiments to be more conservative (see Fig. S15-16). All figures with flow cytometry data have been updated to reflect new gating.

We anticipate the presentation of more detailed gating scheme will clarify any confusion with the CD11b and F480 populations and have included the FMOs below.

FMO missing CD11b/BV510

FMO missing F4/80 / PECy7

F4/80 - PECy7

F4/80 - PECy7

Full Panel

F4/80 - PECy7

REVIEWERS' COMMENTS:

Reviewer #1 (Remarks to the Author):

The authors have taken the advice in consideration and included additional experiments supporting the approach and the conclusions.

Minor comments:

For several citations the journal is not listed, for example:

Miller, S. D., Karpus, W. J. & Davidson, T. S. Experimental Autoimmune Encephalomyelitis in the Mouse. 1–26 (2010).

Please update the reference list and include the journal and correct identifying information.

Reviewer #2 (Remarks to the Author):

The authors has mainly addressed my concerns. The quality of the paper has been improved. There is no any other questions.

REVIEWERS' COMMENTS:

Reviewer #1 For several citations the journal is not listed, for example:

Miller, S. D., Karpus, W. J. & Davidson, T. S. Experimental Autoimmune Encephalomyelitis in the Mouse. 1–26 (2010).

We have updated the references.